# LEVERAGING KV SIMILARITY FOR ONLINE STRUCTURED PRUNING IN LLMS

## ABSTRACT

Pruning has emerged as a promising direction for accelerating large language model (LLM) inference, yet existing approaches often suffer from instability because they rely on offline calibration data that may not generalize across inputs. In this work, we introduce ***Token Filtering***, a lightweight online structured pruning technique that makes pruning decisions directly during inference without any calibration data. The key idea is to measure token redundancy via joint key–value similarity and skip redundant attention computations, thereby reducing inference cost while preserving critical information. To further enhance stability, we design a variance-aware fusion strategy that adaptively weights key and value similarity across heads, ensuring that informative tokens are retained even under high pruning ratios. This design introduces no additional memory overhead and provides a more reliable criterion for token importance. Extensive experiments on LLaMA-2 (7B/13B), LLaMA-3 (8B), and Mistral (7B) demonstrate that Token Filtering consistently outperforms prior structured pruning methods, preserving accuracy on commonsense reasoning benchmarks and maintaining strong performance on challenging tasks such as MMLU, even with 50% pruning.

## 1 INTRODUCTION

Large Language Models (LLMs) (Vaswani et al., 2017; Touvron et al., 2023) have achieved remarkable success across a wide range of tasks, including natural language understanding, reasoning, and generation, and they now serve as the foundation for many state-of-the-art AI applications (OpenAI, 2023). However, their deployment in real-world scenarios remains challenging due to the models' highly complex architectures and massive parameter counts, which result in substantial inference latency and considerable resource consumption.

Pruning is a widely studied technique for accelerating neural networks. Unstructured pruning (Frantar & Alistarh, 2023) adaptively removes individual weights and achieves high compression with modest accuracy loss, but practical speedups often require specialized hardware. Structured pruning (Ma et al., 2023) removes larger components such as attention heads or modules, which is more hardware-friendly but typically causes non-trivial accuracy degradation.

While effective in certain settings, most pruning methods are applied offline using a calibration dataset, which can lead to overfitting and reduced generalization to downstream tasks (Williams & Aletras, 2023). To overcome these limitations, online pruning has emerged as a promising alternative, making pruning decisions dynamically during inference based on real inputs. Unlike offline approaches, it cannot rely on global profiling with calibration data and must instead operate adaptively on local features at runtime. This design presents new challenges: the absence of global saliency information and the need for extremely lightweight decision mechanisms, since any additional computation directly increases inference latency.

Recently, token pruning has emerged as a complementary strategy that directly reduces the sequence length by discarding tokens deemed less informative during inference. By shortening the effective context, token pruning alleviates the quadratic complexity of self-attention and yields substantial reductions in FLOPs and latency. Learned Token Pruning (LTP) (Kim et al., 2022) adaptively drops tokens based on learned attention thresholds, while Zero-TPrune (Wang et al., 2024) leverages attention graphs of pre-trained models to enable zero-shot pruning without retraining. More recently, LazyLLM (Fu et al., 2024) applied token pruning to large language models and achieved over 2×

speedup in long-context inference, but such methods still rely on computing attention scores to estimate token importance, which reduces the potential benefits of pruning by adding extra computation.

In this work, we propose ***Token Filtering***, an online dynamic structured pruning technique that directly reduces inference cost by filtering out redundant tokens in real time and skipping their attention computations. Unlike prior token pruning methods that rely on attention scores to estimate token importance, our approach leverages ***key–value similarity*** as a lightweight redundancy signal, thereby avoiding the overhead of score computation. The key idea is that tokens highly similar to past context are unlikely to contribute novel information and can thus be pruned without harming accuracy. To quantify this redundancy, Token Filtering uses both key similarity and value similarity, defined as the cosine similarity between the current key or value and the mean representation of all previous tokens. In multi-head attention, where different heads capture diverse perspectives, we compute similarity for each head individually and then average them to preserve diversity. To further improve stability, we incorporate a variance-aware fusion strategy: even when the mean similarity is high, a large variance across heads may indicate that some heads still encode important information. We therefore assign greater weight to the feature (key or value) with lower variance and combine them to produce a final similarity score. This similarity-based filtering enables Token Filtering to maintain accuracy under high pruning ratios while eliminating redundant attention operations, all without requiring global profiling or calibration data.

To reduce overhead, we propose a tail-focused pruning strategy. When attention operations are skipped through similarity-based decisions, latency can be significantly reduced; however, when skipping does not occur, the decision-making cost remains as pure overhead. Thus, to ensure sufficient latency gains, it is more effective to apply our method selectively to layers with higher pruning likelihood rather than uniformly across the entire model. Based on empirical observations, we find that later layers, where attention scores are more concentrated on a few tokens, exhibit a higher probability of being pruned. Finally, to maintain the target pruning ratio for each layer, we introduce dynamically adjusted layer-wise thresholds, which are updated based on the current skip ratio. Together, these innovations establish Token Filtering as an effective approach for accelerating LLM inference while preserving accuracy on challenging benchmarks such as commonsense reasoning and MMLU (Hendrycks et al., 2020)

## 2 RELATED WORK

**Pruning.** Network pruning is an effective model compression technique. Pruning methods can be broadly categorized into two types: unstructured pruning and structured pruning (Cheng et al., 2024). Unstructured pruning (Yang et al., 2022; Frantar et al., 2022; Diao et al., 2023) focuses on individual weights, whereas structured pruning (Ma et al., 2023; Le et al., 2025; Ling et al., 2024; An et al., 2024) removes relatively larger structures such as channels, heads, or layers. For unstructured pruning, SparseGPT (Frantar & Alistarh, 2023) addresses the layer-wise pruning problem by employing the Optimal Brain Surgeon (OBS) approach to approximate the Hessian and thus determine saliency. BESA (Xu et al., 2024) applies different pruning ratios per layer, adapting the sparsity allocation across the network. For structured pruning, SlimGPT (Ling et al., 2024) gradually increases the pruning ratio while removing attention heads. FLAP (An et al., 2024) leverages a fluctuation-based pruning metric to adaptively search for a suitable global compression ratio. These methods typically rely on an offline pruning process with a calibration dataset, and since pruning is performed without considering the input at runtime, they often suffer from significant accuracy degradation. More recently, Probe Pruning (Le et al., 2025) demonstrates that probing a small portion of each batch can effectively identify crucial weights, enabling online and dynamic pruning. However, it still requires a calibration dataset to construct historical states, limiting its general applicability. In contrast, our approach performs pruning entirely online, jointly considering both tokens and layers, and achieves this without any calibration dataset, thereby minimizing performance degradation.

**Similarity.** Computation reuse has been studied across different contexts. Mercury (Janfaza et al., 2023) caches intermediate results and reuses them for similar inputs, which requires allocating additional memory storage. In large language models, Prompt Cache (Gim et al., 2024) accelerates inference by storing frequently used prompts, while Key-Diff (Park et al., 2025) compresses the KV cache by exploiting cosine similarity between keys. These approaches primarily target memory reuse or KV-cache-specific optimizations. In contrast, we leverage similarity to skip attention

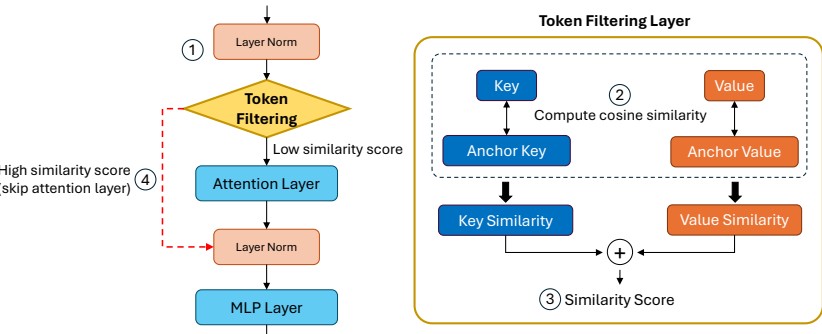

Figure 1: Technique of Token Filtering. ① Tokens first pass through the Token Filtering layer before entering the attention layer. ② Cosine similarity is computed between the key/value and the anchor key/value, where the anchor key/value represents the average of previous keys and values. ③ The key similarity and value similarity are added to obtain the similarity score. ④ If the similarity score is high, the attention layer is skipped.

computations at runtime directly, enabling structured pruning throughout the decoding process. This shift in similarity-based reuse from a cache-oriented paradigm to a compute-oriented pruning mechanism offers broader applicability and stronger efficiency gains.

## 3 METHOD

This section introduces Token Filtering, our method for layer-wise and token-wise online structured pruning in LLMs. The goal is to reduce the substantial inference cost of LLMs while maintaining accuracy at high sparsity levels. Our method has two key components: (i) a lightweight KV-similarity–based selection that jointly considers key and value to identify highly redundant tokens and prune them in real time; (ii) Tail-focused pruning with layer-wise thresholds to minimize latency overhead. An overview of the proposed pruning technique is illustrated in Figure 1.

### 3.1 KV SIMILARITY AND ATTENTION SCORE

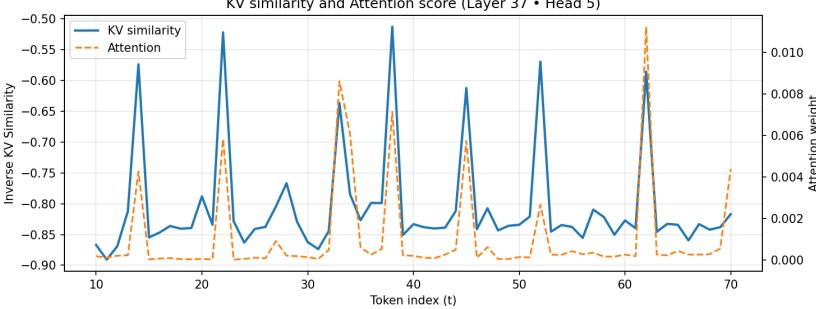

Figure 2: Visualization of inverse KV similarity and attention weights for a representative head (Layer 37, Head 5). For clarity, we plot the inverse cosine similarity so that higher values correspond to lower similarity. Peaks in inverse similarity generally coincide with peaks in attention weights, illustrating that tokens with lower similarity tend to receive higher attention.

As discussed above, Token Filtering performs online pruning during runtime to improve inference speed while preserving accuracy. Offline pruning can exploit metrics obtained from end-to-end profiling, such as attention scores or cosine similarity between inputs and outputs, to determine which components to prune. In contrast, online pruning must operate during inference, which makes

---

**Algorithm 1** Token Filtering Layer Guided Skip Decision

---

**Require:** layer index $i$, input keys $\mathbf{K}$, input values $\mathbf{V}$
    per-layer anchors $\text{ANCHORK}[i]$, $\text{ANCHORV}[i]$, thresholds $\tau[i]$
    running variances $\sigma_k^2[i]$, $\sigma_v^2[i]$
**Require:** global: target skip ratio $\rho^\star$, current skip ratio $\rho$, step size $\eta$

  **function** SKIPDECISION($i, \mathbf{K}, \mathbf{V}$)
      $\text{ANCHORK}[i] \leftarrow \text{mean}(\mathbf{K})$              ▷ anchor key = mean of previous keys
      $\text{ANCHORV}[i] \leftarrow \text{mean}(\mathbf{V})$            ▷ anchor value = mean of previous values
      $\mathbf{k}_{\text{last}} \leftarrow \mathbf{K}[:, -1, :]$
      $\mathbf{v}_{\text{last}} \leftarrow \mathbf{V}[:, -1, :]$
      $(s_k, \sigma_k^2[i]) \leftarrow (\text{mean}, \text{var})(\text{cosine\_sim}(\text{ANCHORK}[i], \mathbf{k}_{\text{last}}))$
      $(s_v, \sigma_v^2[i]) \leftarrow (\text{mean}, \text{var})(\text{cosine\_sim}(\text{ANCHORV}[i], \mathbf{v}_{\text{last}}))$
      $\alpha \leftarrow \frac{1/\sigma_v^2[i]}{1/\sigma_k^2[i] + 1/\sigma_v^2[i]}$         ▷ variance-based weighting
      $s_{\text{KV}} \leftarrow \alpha \, s_k + (1 - \alpha) \, s_v$         ▷ final similarity score
      $\tau[i] \leftarrow \tau[i] + \eta(\rho^\star - \rho)$          ▷ threshold adaptation
      **Skip layer if** $s_{\text{KV}} > \tau[i]$
  **end function**

---

it infeasible to rely on such global statistics. Therefore, a new criterion is required, and we draw inspiration from input similarity to guide pruning decisions.

The relationship between keys and attention scores has been discussed in many prior studies. KeyDiff (Park et al., 2025) demonstrated a correlation between the cosine similarity among keys and their corresponding attention scores. In particular, the anchor key, defined as the mean of all previous keys, serves as a reference. Keys with high cosine similarity to the anchor key are considered redundant and less important, which allows them to be evicted from the KV cache. However, attention pruning is more aggressive than KV cache eviction, and relying solely on key similarity to assess token importance can be unstable.

Therefore, we also incorporate values into the pruning criterion. Although keys and values are generated from the same token, additional factors, such as positional embeddings applied only to keys, cause them to encode different information. By jointly considering key similarity and value similarity, we obtain a more stable and reliable measure of token importance. Figure 2 illustrates the relationship between the variance-aware KV similarity, introduced in the following subsection, and the attention scores. For clarity of visualization, we plot the inverse KV similarity. We observe an inverse relationship, where tokens with higher similarity tend to receive lower attention scores. This indicates that similarity can serve as a reliable predictor of less important tokens, allowing us to anticipate redundant attention computations without explicitly performing them.

## 3.2 TOKEN FILTERING

By leveraging key and value similarity, we can prune redundant tokens before performing the attention operation. To maximize this benefit, we insert an additional Token Filtering layer into each Transformer block, placing it before the attention layer. The Token Filtering layer computes the key and value similarity and provides a skip path that bypasses the attention computation. The overall procedure of the Token Filtering layer is summarized in Algorithm 1.

Even when a token is skipped, LayerNorm is still computed as usual, and the attention output for that token is replaced with zero. Consequently, after passing through the residual connection, the block's output becomes identical to its input. This prevents Token Filtering from effectively changing the structure of the Transformer, even when attention computations are skipped.

In the Token Filtering layer, we define KV similarity as the importance score used to decide whether a token should be pruned. Specifically, KV similarity is computed as a weighted combination of key similarity and value similarity. Key and value similarity are defined as the cosine similarity between the anchor key or value and the current token's key or value. In multi-head attention, keys and values differ across heads. If we first average keys and values across the head dimension and

then compute cosine similarity, the computational cost can be reduced, but this approach discards head-specific information and prevents the estimation of variance across heads. To preserve head-level information, we instead compute cosine similarity for each head individually and then take the average to obtain the final key and value similarities.

The anchor key is defined as the mean of all previous keys. However, computing head-wise cosine similarity introduces additional overhead. To mitigate this cost, we replace the exact average with an efficient incremental averaging strategy, where a fixed smoothing factor $\gamma$ is used to retain the influence of recent tokens:

$$\text{Anchor} \leftarrow \gamma \cdot \text{Anchor} + (1 - \gamma) \cdot \mathbf{k}_{\text{current}}, \tag{1}$$

where $\gamma$ is close to one (e.g., $\gamma = 0.9$). Using a fixed factor emphasizes more recent inputs and avoids the dominance of older tokens, effectively capturing a recent window of key information. The same update rule is applied to anchor values.

Because each layer maintains only one anchor key and one anchor value (one vector per head), the memory footprint is extremely small. The total anchor size is approximately 800 KB, which is negligible compared to the KV cache. For reference, the KV cache typically occupies several hundred megabytes depending on the batch size and sequence length, making the anchor overhead effectively zero.

Since we compute similarity for each head individually, we can also obtain the variance of similarity across heads. In multi-head attention, each head represents a different perspective, meaning that the keys and values of different heads correspond to distinct interpretations of the same token. Thus, head-wise similarity can be regarded as measuring token redundancy from multiple viewpoints. Leveraging variance allows us to avoid the pitfalls of averaging: even if the mean similarity is high, a significant variance may indicate that some heads capture non-redundant, and therefore necessary, information. Consequently, when combining key and value similarities, we assign greater weight to the side with lower variance, reflecting higher consistency across heads. Formally, let $s_k$ and $s_v$ denote the cosine similarities of key and value, and let $\sigma_k^2$ and $\sigma_v^2$ be their running variances. The final similarity score is given by:

$$\alpha = \frac{\sigma_v^{-2}}{\sigma_k^{-2} + \sigma_v^{-2}}, \tag{2}$$

$$S_{\text{KV}} = \alpha \, s_k + (1 - \alpha) \, s_v. \tag{3}$$

Here, $\alpha$ is adaptively determined for each head, so that the more stable feature (lower variance) contributes more to the final similarity score. This variance-aware weighting yields a more robust criterion for token pruning than relying solely on key similarity.

To prevent unstable pruning decisions at the beginning of decoding, we introduce a short warm-up phase. During this warm-up period, the layer-wise thresholds are updated, but no pruning is performed. This ensures that pruning is triggered only after the anchor keys and values have accumulated sufficient historical information, avoiding cold-start instability.

### 3.3 TAIL-FOCUSED PRUNING WITH LAYER-WISE THRESHOLD

The goal of Token Filtering is to achieve acceleration comparable to offline pruning, despite operating in an online setting. Unlike offline pruning, which behaves like a dense model at inference, online pruning inevitably exposes the computational overhead required for pruning decisions. To address this, we perform the additional pruning computations only in layers with a high likelihood of being pruned, rather than applying them uniformly across all layers. Many previous studies have investigated which layers are less critical and can be pruned with minimal impact on model performance. For example, (Gao et al., 2025) estimates the pruning sensitivity of each layer and performs non-uniform pruning accordingly. Similarly, (Lu et al., 2024) empirically investigates various layer pruning strategies and adopts a heuristic of removing the final 25% of layers.

Based on prior studies and our empirical observation, we adopt a tail-focused online structured pruning strategy. Instead of pruning every layer uniformly, we prune only the later layers. Concretely, to meet a global pruning budget $P_{global}$ (e.g., 25%), we select the last Y fraction of layers (e.g., Y=0.5) and prune them more aggressively so that the per-selected-layer target becomes:

$$P_{\text{tail}} = \frac{P_{\text{global}}}{Y}. \tag{4}$$

Table 1: Zero-shot evaluation of LLaMA-2-13B on perplexity (PPL, ↓) and commonsense reasoning benchmarks. The "Avg." column reports the average accuracy across the seven tasks. The **bolded** results indicate the best result within each pruning ratio group.

| Prune% | Method | PPL ↓ | BoolQ | PIQA | HellaS | WinoG | ARC-e | ARC-c | OBQA | Avg. |
|---|---|---|---|---|---|---|---|---|---|---|
| 0% | Dense | 10.98 | 80.92 | 80.52 | 79.36 | 71.98 | 79.63 | 49.15 | 45.00 | 69.51 |
| 20% | SlimGPT w/o | 13.80 | **80.37** | 78.45 | 77.07 | 71.51 | 75.84 | 45.14 | 43.60 | 67.43 |
| | FLAP | 14.13 | 77.21 | 77.95 | **78.06** | 71.32 | 76.44 | 45.27 | 42.00 | 66.89 |
| | PP | **12.52** | 76.29 | **79.55** | 76.53 | 68.57 | 76.85 | 44.57 | 42.00 | 66.33 |
| | Token Filtering | 13.37 | 80.18 | 78.62 | 77.78 | **72.22** | **78.96** | **48.04** | **45.00** | **68.69** |
| 25% | SlimGPT w/o | 15.10 | 78.78 | 77.91 | 75.65 | 70.64 | 73.06 | 44.11 | 43.00 | 66.16 |
| | FLAP | 15.49 | 75.81 | 75.61 | 74.74 | 69.59 | 73.83 | 43.57 | 41.00 | 64.88 |
| | PP | **13.32** | 73.82 | 78.47 | 75.82 | 67.78 | 75.42 | 42.95 | 42.40 | 65.24 |
| | Token Filtering | 14.69 | **80.09** | **78.56** | **77.82** | **71.03** | **77.57** | **47.95** | **44.20** | **68.17** |
| 33% | SlimGPT w/o | 18.11 | 76.27 | 76.44 | 72.76 | 70.80 | 70.83 | 41.21 | 41.00 | 64.19 |
| | FLAP | 17.79 | 74.37 | 74.42 | 70.45 | 68.17 | 70.22 | 42.29 | 39.20 | 62.73 |
| | PP | **15.83** | 70.30 | 76.74 | 71.67 | 63.65 | 71.08 | 39.96 | 42.60 | 62.29 |
| | Token Filtering | 16.39 | **79.79** | **77.15** | **76.07** | **71.35** | **77.44** | **47.35** | **44.00** | **67.65** |
| 50% | SlimGPT w/o | 32.67 | 66.06 | 73.61 | 61.76 | 65.82 | 60.44 | 34.39 | 38.00 | 57.15 |
| | FLAP | 29.45 | 74.31 | 70.49 | 58.39 | 62.96 | 61.67 | 36.83 | 37.20 | 57.41 |
| | PP | **28.86** | 62.17 | 69.22 | 49.88 | 55.07 | 59.26 | 29.63 | 36.20 | 51.93 |
| | Token Filtering | 29.22 | **79.76** | **77.04** | **74.56** | **71.03** | **71.72** | **45.22** | **42.00** | **65.90** |

Each selected layer $l$ maintains its own learnable threshold $T_l$. At each decoding iteration, each selected layer $l$ computes its own layer-wise similarity score $S_{KV}^{(l)}$ (from Eq. (2)). If $S_{KV}^{(l)} > T_l$, we skip the attention of that layer for the current tokens (FFN is still executed). Let $P_{current}$ be the observed current skip ratio of layer $l$ (running estimate), which denotes the proportion of times this layer has been skipped so far during inference. We update $t_l$ online via proportional feedback to match $P_{tail}$:

$$T_l \leftarrow T_l + \eta(P_{\text{current}} - P_{\text{tail}}). \tag{5}$$

where $\eta > 0$ is a small learning rate. Layers outside the tail set are not pruned (no skipping). This concentrates pruning where layers are typically less sensitive, while keeping the global skip budget close to $P_{global}$.

## 4 EXPERIMENT

### 4.1 EXPERIMENTAL SETTINGS

**Setup.** We conduct all experiments on a single NVIDIA A100 GPU with 80GB of memory. All LLM models and datasets are obtained from the Hugging Face Transformers library (Wolf et al., 2020). Zero-shot evaluations on commonsense reasoning benchmarks and the MMLU benchmark are performed using the lm-eval-harness framework (Gao et al., 2021). For all baseline methods, we used the official implementations released by the original authors.

**Models.** Our evaluation primarily targets a diverse set of LLMs to demonstrate the generality and robustness of our approach. Specifically, we experiment with models from the LLaMA family, including LLaMA-2 7B/13B (Touvron et al., 2023) and LLaMA-3 8B (Meta AI, 2024), as well as the Mistral-7B (Jiang et al., 2023) and Phi-4-14B (Abdin et al., 2024) models. By covering various model families and a wide range of parameter scales, we show that our methodology consistently yields improvements and is not restricted to a specific model type or size.

**Benchmarks.** To evaluate the effectiveness of Token Filtering, we measure both perplexity and accuracy following prior pruning studies. For language modeling performance, we report perplexity on the WikiText2 (Merity et al., 2016) validation set with sequence length truncated to 128. For com-

Table 2: Zero-shot evaluation of various models on text generation (perplexity, ↓) and commonsense reasoning (accuracy, ↑). A dash (−) indicates that the baseline method could not be applied successfully to the corresponding model.

| Method | Pruning Ratio | Text Generation ↓ | | | | Commonsense Reasoning ↑ | | | |
|---|---|---|---|---|---|---|---|---|---|
| | | LLaMA-2-7B | LLaMA-3-8B | Mistral-7B | Phi-4-14B | LLaMA-2-7B | LLaMA-3-8B | Mistral-7B | Phi-4-14B |
| Dense | 0% | 12.18 | 14.13 | 11.90 | 16.20 | 66.88 | 70.39 | 71.40 | 72.66 |
| 20% | SlimGPT w/o | 16.36 | 32.79 | 20.39 | 191.31 | **64.27** | 62.76 | 55.51 | 47.61 |
| | FLAP | 16.32 | 23.25 | 16.08 | - | 63.10 | 55.75 | 57.72 | - |
| | PP | **15.31** | 20.56 | - | - | 63.34 | 58.66 | - | - |
| | Token Filtering | 16.65 | **19.46** | **15.52** | **20.52** | 63.61 | **67.94** | **65.82** | **70.52** |
| 50% | SlimGPT w/o | 40.83 | 85.32 | 43.68 | 512.32 | 51.45 | 47.68 | 43.20 | 32.53 |
| | FLAP | 43.11 | **63.33** | **42.02** | - | 46.97 | 42.61 | 44.57 | - |
| | PP | **39.40** | 104.01 | - | - | 50.59 | 41.54 | - | - |
| | Token Filtering | 54.59 | 74.62 | 48.08 | **72.89** | **53.13** | **59.31** | **50.37** | **59.87** |

Table 3: MMLU zero-shot performance of LLaMA-2-7B/13B. Here, "social" denotes the social sciences category. Probe Pruning (PP) is excluded since its official implementation does not support MMLU evaluation.

| Method | Pruning Ratio | LLaMA-2-7B | | | | | LLaMA-2-13B | | | | |
|---|---|---|---|---|---|---|---|---|---|---|---|
| | | Humanities | Social | STEM | Other | Avg | Humanities | Social | STEM | Other | Avg |
| 0% | Dense | 39.21 | 45.99 | 33.17 | 45.41 | 40.81 | 47.89 | 61.03 | 42.44 | 59.29 | 52.07 |
| 20% | SlimGPT | 30.47 | 30.48 | 26.07 | 34.37 | 30.35 | 43.38 | 54.01 | 39.30 | 52.98 | 46.92 |
| | FLAP | 28.01 | 33.67 | 30.23 | 31.89 | 30.61 | 42.77 | 53.92 | 40.46 | 47.66 | 45.79 |
| | Token Filtering | **36.03** | **41.18** | **32.25** | **42.68** | **37.78** | **47.06** | **59.86** | **41.80** | **58.22** | **51.15** |
| 50% | SlimGPT | 23.44 | 22.23 | 21.33 | 23.82 | 22.96 | 31.56 | 30.39 | 27.56 | 31.57 | 30.41 |
| | FLAP | 24.21 | 21.71 | 21.25 | 23.98 | 22.95 | 34.29 | 31.84 | 28.57 | 30.14 | 30.81 |
| | Token Filtering | **33.84** | **38.45** | **30.63** | **39.17** | **35.31** | **44.23** | **53.23** | **38.63** | **52.08** | **46.68** |

monsense reasoning capabilities, we conduct zero-shot evaluations on seven standard benchmarks: BoolQ (Clark et al., 2019), PIQA (Bisk et al., 2020), HellaSwag (Zellers et al., 2019), WinoGrande (Sakaguchi et al., 2021), ARC-Easy (Clark et al., 2018), ARC-Challenge (Clark et al., 2018), and OpenBookQA (Mihaylov et al., 2018). Additionally, we evaluate on MMLU, a knowledge-intensive benchmark that encompasses 57 subjects requiring domain-specific reasoning.

**Baselines.** We evaluate the following established methods: (i) SlimGPT (Ling et al., 2024), an inference-time pruning method based on the Optimal Brain Surgeon (OBS) framework that progressively prunes attention heads. (ii) FLAP (An et al., 2024), a retraining-free structured pruning framework that uses input feature fluctuations on a calibration set to search for global compression structures. (iii) Probe Pruning (PP) (Le et al., 2025), an online dynamic pruning method that probes a small subset of tokens to guide batch-wise channel pruning during inference.

## 4.2 MAIN RESULT

### 4.2.1 ACCURACY AND PERPLEXITY RESULTS

In this experimental evaluation, we report the zero-shot performance of pruned models without any fine-tuning on both text generation and commonsense reasoning tasks. Table 1 presents the detailed perplexity and accuracy results of the LLaMA-2-13B model under various pruning ratios. Compared to other approaches, Token Filtering achieves superior performance across most subtasks. Under a pruning ratio of 20%, Token Filtering shows only marginal gains, with a perplexity higher than the best existing result (13.37 vs. 12.52) but an accuracy improvement of 1.23 points (68.69 vs. 67.43). However, as the pruning ratio increases, the advantage of Token Filtering becomes evident. At 50% pruning, Token Filtering achieves a comparable perplexity (29.22 vs. 28.86) and a substantial accuracy gain of 8.49 points (65.90 vs. 57.41) over the best baseline.

Table 2 reports zero-shot evaluation results on multiple LLM families under different pruning ratios. Overall, Token Filtering achieves comparable perplexity while delivering superior accuracy across most settings. On Mistral-7B and LLaMA-3-8B, baseline methods exhibit rapid performance degra-

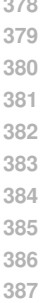

Figure 3: Latency and memory reduction (%) of Token Filtering under a 50% pruning ratio across different batch sizes on LLaMA-2-13B

Table 4: Attention share in total latency at different batch sizes.

| Batch size | Total latency (s) | Attention latency (s) | Attention share (%) |
|---|---|---|---|
| 8 | 1.9 | 1.3 | 68% |
| 128 | 24.1 | 23.3 | 97% |

dation even at low pruning ratios, whereas Token Filtering maintains strong performance under the same conditions. For the latest Phi-4-14B model, most prior methods either fail to run or suffer from severe model collapse, while our approach sustains a reasonable level of performance. On smaller 7B-scale models, all pruning techniques, including Token Filtering, show substantial accuracy drops under high pruning ratios, reflecting the limited redundancy of compact models. More detailed results and breakdowns across subtasks are provided in the Appendix.

In contrast to prior studies, which reported that aggressively pruned large models (e.g., LLaMA-13B at 50% sparsity) may underperform lightly pruned smaller counterparts (e.g., LLaMA-7B at 20% sparsity) due to limited recovery under low-cost fine-tuning, our method demonstrates stronger robustness. Specifically, our 50% pruned LLaMA-2-13B achieves zero-shot accuracy comparable to the dense LLaMA-2-7B (65.90 vs. 66.88), indicating that our approach effectively preserves the representational capacity of large models even under high pruning ratios. This result highlights the advantage of our online pruning strategy, which can sustain high pruning levels without sacrificing task performance, thereby offering more favorable efficiency–accuracy trade-offs than conventional offline pruning approaches.

We further evaluate the pruned models on MMLU, a challenging benchmark that assesses broad multi-domain knowledge and reasoning capabilities. As shown in Table 3, Token Filtering consistently outperforms prior structured pruning methods by a large margin. For LLaMA-2-7B at a 20% pruning ratio, Token Filtering achieves an average accuracy of 37.78%, substantially higher than SlimGPT (30.35%) and FLAP (30.61%). At a 50% pruning ratio, the gap becomes even more pronounced: Token Filtering attains 35.31%, compared to only 22.96% for SlimGPT and 22.95% for FLAP. A similar trend holds for the larger LLaMA-2-14B model, where Token Filtering maintains 51.15% accuracy at 20% pruning and 46.68% at 50%, while competing methods collapse to below 47% and 31%, respectively. These results highlight the robustness of Token Filtering on complex reasoning tasks, demonstrating its ability to preserve knowledge-intensive performance even under aggressive pruning.

### 4.2.2 EFFICIENCY EVALUATION

Figure 3 presents the end-to-end latency and memory reduction achieved by Token Filtering under a 50% pruning ratio across different batch sizes. The results are obtained on the LLaMA-2-13B model with an output limit of 512 tokens, averaging over 10 runs. We use the prompt "Once upon a time in a land far, far away," as the input, which corresponds to 13 tokens under the LLaMA tokenizer. As shown in Figure 3, our method reduces both latency and memory across different batch sizes.

Table 5: End-to-end throughput (tokens/s) measured on LLaMA-2-13B across different batch sizes (8–128) under a 50% pruning ratio.

| Prune% | Method | 8 batch | 16 batch | 32 batch | 64 batch | 128 batch |
|--------|--------|---------|----------|----------|----------|-----------|
| 0% | Dense (token/s) | 16.8 | 16.1 | 11.8 | 4.7 | 2.7 |
| 50% | Token Filtering (token/s) | 19.4 | 17.8 | 14.9 | 8.8 | 5.2 |
| | FLAP (token/s) | 19.4 | 19.1 | 15.4 | 9.6 | 5.8 |
| | Probe (token/s) | 19.2 | 17.2 | 14.6 | 7.5 | 4.4 |

Table 6: Comparison of pruning focus strategies (uniform, head-focused, and tail-focused) on LLaMA-2-13B. Results are reported for perplexity (PPL, ↓) and seven commonsense reasoning benchmarks.

| Prune% | Method | PPL↓ | BoolQ | PIQA | HellaS | WinoG | ARC-e | ARC-c | OBQA | Avg. |
|--------|--------|------|-------|------|--------|-------|-------|-------|------|------|
| 0% | Dense | 10.98 | 80.92 | 80.52 | 79.36 | 71.98 | 79.63 | 49.15 | 45.00 | 69.51 |
| 20% | Uniform | 43.19 | 68.07 | 69.91 | 56.40 | 60.30 | 62.46 | 41.38 | 38.20 | 56.67 |
| | Head-focused | 179.02 | 57.68 | 57.78 | 37.60 | 55.09 | 43.81 | 30.55 | 32.40 | 44.99 |
| | Tail-focused | **13.37** | **80.18** | **78.62** | **77.78** | **72.22** | **78.96** | **48.04** | **45.00** | **68.69** |
| 50% | Uniform | 429.44 | 51.83 | 52.39 | 29.82 | 51.70 | 35.10 | 25.34 | 31.00 | 39.60 |
| | Head-focused | 6717.69 | 37.83 | 50.82 | 26.55 | 47.67 | 26.52 | 28.24 | 25.80 | 34.78 |
| | Tail-focused | **29.22** | **79.76** | **77.04** | **74.56** | **71.03** | **71.72** | **45.22** | **42.00** | **65.90** |

Notably, the reduction becomes more pronounced as the batch size increases: at a batch size of 128, latency is reduced by 46.6% and memory usage by 33.6%.

Interestingly, our measurements show that the latency of the MLP and normalization layers remains nearly constant, regardless of the batch size. In contrast, the latency of the attention layer increases sharply with larger batches. As shown in Table 4, at batch size 128, attention accounts for 23 seconds out of the 24-second total latency (≈97%). This behavior stems from the inherently non-parallelizable nature of attention operations and explains the batch-dependent variation in latency reductions observed in Figure 3. Since Token Filtering directly prunes attention layers, it achieves nearly half the latency reduction at large batch sizes, where attention dominates the runtime.

We also observe that this batch-dependent behavior is not unique to Token Filtering. Other pruning-based acceleration methods (e.g., FLAP, Probe Pruning) show a similar trend where performance gains are limited at small batch sizes and become increasingly pronounced as the batch size grows.

To provide a direct latency comparison with existing pruning baselines, we report the throughput (tokens per second) of Token Filtering, FLAP, and Probe Pruning under a 50% pruning ratio across various batch sizes in Table 5. Token Filtering achieves slightly lower throughput than FLAP, an offline pruning method with no runtime overhead, but it consistently surpasses Probe Pruning, which is an online pruning method like ours.

## 4.3 EXTENDED ANALYSIS

### 4.3.1 PRUNING FOCUS: TAIL, HEAD, AND UNIFORM

Table 6 compares uniform, head-focused, and tail-focused pruning strategies under both 20% and 50% pruning ratios. The results clearly show that tail-focused pruning consistently outperforms the other approaches in both perplexity and accuracy. At 20% pruning, tail-focused Token Filtering achieves a perplexity of 13.37 and an average accuracy of 68.69, significantly better than uniform pruning (PPL 43.19, Avg 56.67) and head-focused pruning (PPL 179.02, Avg 44.99). A similar trend is observed at 50% pruning, where tail-focused pruning maintains competitive accuracy (65.90) with substantially lower perplexity (29.22) compared to the uniform (PPL 429.44, Avg 39.60) and head-focused (PPL 6717.69, Avg 34.78) variants. These results demonstrate that pruning later layers

Table 7: Comparison of importance criteria (key-only, value-only, and KV) on LLaMA-2-7B and LLaMA-2-13B under a 33% pruning ratio. Results are reported for perplexity (PPL, ↓), commonsense reasoning accuracy, and MMLU accuracy.

| Pruning% | Method | LLaMA-2-7B | | | LLaMA-2-13B | | |
|---|---|---|---|---|---|---|---|
| | | PPL↓ | Commonsense Reasoning | MMLU | PPL↓ | Commonsense Reasoning | MMLU |
| 33% | Key-only | **24.13** | 58.98 | 35.11 | **16.39** | 67.06 | 50.14 |
| | Value-only | 24.85 | **59.54** | 35.08 | 16.65 | 67.64 | 48.56 |
| | KV | 24.22 | 59.15 | **35.51** | **16.39** | **67.65** | **49.48** |

Table 8: Needle-in-a-Haystack (NiH) retrieval accuracy (%) under 20% pruning on LLaMA-2-13B across different input token length (512-3072).

| Prune% | Method | 512 | 1024 | 1536 | 2048 | 2560 | 3072 |
|---|---|---|---|---|---|---|---|
| 0% | Dense | 100 | 100 | 90 | 95 | 45 | 35 |
| 20% | Token Filtering | 70 | 70 | 65 | 55 | 30 | 10 |
| | FLAP | 25 | 20 | 10 | 0 | 0 | 0 |

is more effective, as it reduces redundancy while minimizing accuracy degradation, making tail-focused pruning the most reliable strategy across different pruning ratios.

### 4.3.2 SIMILARITY: KEY, VALUE, AND KV

Table 7 compares pruning criteria based on key similarity, value similarity, and their combination (KV) under a 33% pruning ratio. The results indicate that perplexity and commonsense reasoning accuracy remain broadly comparable across the three variants, suggesting that either keys or values alone are sufficient for relatively simple reasoning tasks. However, on the more challenging MMLU benchmark, KV consistently achieves the highest performance, reaching 35.51 on LLaMA-2-7B and 49.48 on LLaMA-2-13B, outperforming both key-only and value-only pruning. This finding suggests that while commonsense reasoning tasks may not strongly depend on the integration of key and value information, complex knowledge-intensive evaluations such as MMLU benefit from the added robustness of combining both features.

### 4.3.3 LONG-CONTEXT BENCHMARK

To evaluate the long-context robustness of Token Filtering, we conduct the Needle-in-a-Haystack (NiH) benchmark across six context lengths: 512, 1024, 1536, 2048, 2560, and 3072 tokens. For each context length, we insert the "needle" sentence at 20 uniformly distributed depth positions (from 1% to 99% of the document) and measure the retrieval success rate.

Table 8 summarizes the Needle-in-a-Haystack (NiH) retrieval accuracy under a 20% pruning ratio on LLaMA-2-13B, comparing Token Filtering against the dense model and prior pruning methods. Token Filtering maintains substantially higher retrieval success than FLAP across all context lengths and preserves long-range dependency information much more effectively, especially beyond 1500 tokens where baseline pruning methods sharply degrade.

## 5 CONCLUSION

In this work, we propose Token Filtering, a fully online, dynamic, structured pruning technique for large language models. To address the challenges of online structured pruning, we leveraged joint key–value similarity as a lightweight importance criterion and introduced a tail-focused mechanism to mitigate performance degradation. Extensive experiments demonstrate that Token Filtering can effectively compress diverse models at various pruning ratios without requiring any calibration data or fine-tuning, while consistently preserving accuracy.

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

# A    MORE DETAILED EVALUATION RESULTS

In this section, we present the detailed results of the evaluation for each task. Table 9, 10, 11, and 12 present the detailed perplexity and accuracy results under various pruning ratios. Table 13, 14, and 15 present the evaluation results for MMLU.

Table 9: Zero-shot evaluation of LLaMA-2-7B on perplexity (PPL, ↓) and commonsense reasoning benchmarks. The "Avg." column reports the average accuracy across the seven tasks. The **bolded** results indicate the best result within each pruning ratio group.

| Prune% | Method | PPL ↓ | BoolQ | PIQA | HellaS | WinoG | ARC-e | ARC-c | OBQA | Avg. |
|--------|--------|-------|-------|------|--------|-------|-------|-------|------|------|
| 0% | Dense | 12.18 | 77.77 | 78.67 | 75.98 | 68.82 | 76.35 | 46.16 | 44.40 | 66.88 |
| 20% | SlimGPT w/o | 16.36 | **73.88** | **77.80** | 72.93 | 68.03 | **73.82** | 41.81 | **41.60** | **64.27** |
| | FLAP | 16.32 | 75.09 | 74.59 | 70.38 | **68.69** | 71.34 | 42.20 | 39.40 | 63.10 |
| | PP | **15.31** | 70.88 | 76.61 | **73.18** | 66.11 | 73.56 | 41.67 | 41.40 | 63.34 |
| | Token Filtering | 16.65 | 73.82 | 75.90 | 72.27 | 66.77 | 72.73 | **42.75** | 41.00 | 63.61 |
| 50% | SlimGPT w/o | 40.83 | 59.72 | 70.53 | 53.71 | 59.55 | 50.37 | 32.48 | 33.80 | 51.45 |
| | FLAP | 43.11 | 55.44 | 62.24 | 46.09 | 59.38 | 41.75 | 31.88 | 32.00 | 46.97 |
| | PP | **39.40** | 56.45 | 70.59 | 60.55 | 51.58 | **59.55** | 29.97 | 35.40 | 50.59 |
| | Token Filtering | 54.59 | **64.46** | 66.81 | **61.03** | **64.72** | 46.93 | **34.13** | 33.80 | **53.13** |

Table 10: Zero-shot evaluation of LLaMA-3-8B on perplexity (PPL, ↓) and commonsense reasoning benchmarks. The "Avg." column reports the average accuracy across the seven tasks. The **bolded** results indicate the best result within each pruning ratio group.

| Prune% | Method | PPL ↓ | BoolQ | PIQA | HellaS | WinoG | ARC-e | ARC-c | OBQA | Avg. |
|--------|--------|-------|-------|------|--------|-------|-------|-------|------|------|
| 0% | Dense | 14.13 | 81.10 | 80.69 | 79.13 | 73.16 | 80.13 | 53.49 | 45.00 | 70.39 |
| 20% | SlimGPT w/o | 32.79 | 74.31 | **77.86** | 66.33 | 66.06 | 73.15 | 41.98 | 39.60 | 62.76 |
| | FLAP | 23.25 | 69.14 | 71.49 | 54.44 | 63.69 | 59.85 | 34.22 | 37.40 | 55.75 |
| | PP | 20.56 | 64.68 | 77.45 | 65.55 | 61.51 | 65.36 | 39.28 | 36.80 | 58.66 |
| | Token Filtering | **19.46** | **80.40** | 77.37 | **74.21** | **73.80** | **76.85** | **50.17** | **42.80** | **67.94** |
| 50% | SlimGPT w/o | 85.32 | 59.83 | 70.12 | 58.21 | 58.14 | 42.57 | 25.31 | 31.60 | 47.68 |
| | FLAP | **63.33** | 57.06 | 58.76 | 36.24 | 54.54 | 38.22 | 24.23 | 29.20 | 42.61 |
| | PP | 104.01 | 50.03 | 69.52 | 29.00 | 53.01 | 36.57 | 23.91 | 28.80 | 41.54 |
| | Token Filtering | 74.62 | **80.28** | **70.13** | **63.37** | **71.03** | **55.56** | **38.82** | **36.00** | **59.31** |

Table 11: Zero-shot evaluation of Mistral-7B on perplexity (PPL, ↓) and commonsense reasoning benchmarks. The "Avg." column reports the average accuracy across the seven tasks. The **bolded** results indicate the best result within each pruning ratio group. A dash (−) indicates that the baseline method could not be applied successfully to the corresponding model.

| Prune% | Method | PPL ↓ | BoolQ | PIQA | HellaS | WinoG | ARC-e | ARC-c | OBQA | Avg. |
|--------|--------|-------|-------|------|--------|-------|-------|-------|------|------|
| 0% | Dense | 11.90 | 83.64 | 82.26 | 81.05 | 73.80 | 80.93 | 53.92 | 44.20 | 71.40 |
| 20% | SlimGPT w/o | 20.39 | 65.81 | 71.76 | 64.67 | 64.40 | 52.95 | 34.39 | 34.60 | 55.51 |
| | FLAP | 16.08 | 68.59 | 68.99 | 58.06 | 65.19 | 67.00 | 37.63 | 38.60 | 57.72 |
| | PP | - | - | - | - | - | - | - | - | - |
| | Token Filtering | **15.52** | **74.98** | **76.61** | **76.64** | **69.30** | **76.14** | **45.73** | **41.40** | **65.82** |
| 50% | SlimGPT w/o | 53.68 | **55.87** | 64.13 | 44.45 | 52.21 | 30.96 | 23.64 | 31.20 | 43.20 |
| | FLAP | 49.12 | 55.24 | 59.11 | 40.52 | 56.16 | 44.81 | 24.77 | 31.40 | 44.57 |
| | PP | - | - | - | - | - | - | - | - | - |
| | Token Filtering | **48.08** | 43.15 | **68.99** | **61.19** | **63.69** | **49.28** | **34.30** | **32.00** | **50.37** |

Table 12: Zero-shot evaluation of Phi-4-14B on perplexity (PPL, ↓) and commonsense reasoning benchmarks. The "Avg." column reports the average accuracy across the seven tasks. The **bolded** results indicate the best result within each pruning ratio group. A dash (−) indicates that the baseline method could not be applied successfully to the corresponding model.

| Prune% | Method | PPL ↓ | BoolQ | PIQA | HellaS | WinoG | ARC-e | ARC-c | OBQA | Avg. |
|---|---|---|---|---|---|---|---|---|---|---|
| 0% | Dense | 16.20 | 86.09 | 81.27 | 82.08 | 76.72 | 81.31 | 56.14 | 45.20 | 72.66 |
| 20% | SlimGPT w/o | 191.31 | 62.20 | 66.54 | 47.52 | 58.96 | 37.96 | 27.30 | 32.80 | 47.61 |
| | FLAP | - | - | - | - | - | - | - | - | - |
| | PP | - | - | - | - | - | - | - | - | - |
| | Token Filtering | **20.52** | **83.00** | **79.43** | **79.72** | **73.24** | **79.21** | **54.27** | **44.80** | **70.52** |
| 50% | SlimGPT w/o | 512.32 | 33.25 | 50.16 | 25.15 | 44.86 | 24.61 | 23.26 | 26.40 | 32.53 |
| | FLAP | - | - | - | - | - | - | - | - | - |
| | PP | - | - | - | - | - | - | - | - | - |
| | Token Filtering | **72.89** | **69.79** | **74.92** | **68.52** | **66.85** | **58.21** | **43.17** | **37.60** | **59.87** |

Table 13: MMLU zero-shot performance of LLaMA-3-8B. Here, "social" denotes the social sciences category. Probe Pruning (PP) is excluded since its official implementation does not support MMLU evaluation.

| Method | Pruning Ratio | LLaMA-3-8B | | | | |
|---|---|---|---|---|---|---|
| | | Humanities | Social | STEM | Other | Avg |
| 0% | Dense | 39.21 | 45.99 | 33.17 | 45.41 | 40.81 |
| 20% | SlimGPT w/o | 43.51 | 54.63 | 40.88 | 52.91 | 47.44 |
| | FLAP | 42.81 | 54.32 | 41.42 | 47.69 | 45.99 |
| | Token Filtering | **45.65** | **56.42** | **43.20** | **59.80** | **50.59** |
| 50% | SlimGPT w/o | 22.68 | 21.57 | 20.32 | 21.41 | 21.01 |
| | FLAP | 24.28 | 21.44 | 20.57 | 22.14 | 22.32 |
| | Token Filtering | **27.55** | **26.91** | **24.99** | **32.19** | **27.86** |

Table 14: MMLU zero-shot performance of Mistral-7B. Here, "social" denotes the social sciences category. Probe Pruning (PP) is excluded since its official implementation does not support MMLU evaluation.

| Method | Pruning Ratio | Mistral-7B | | | | |
|---|---|---|---|---|---|---|
| | | Humanities | Social | STEM | Other | Avg |
| 0% | Dense | 39.21 | 45.99 | 33.17 | 45.41 | 40.81 |
| 20% | SlimGPT w/o | 24.14 | 21.32 | 21.50 | 23.72 | 22.84 |
| | FLAP | 30.09 | 29.57 | 27.05 | 32.63 | 29.86 |
| | Token Filtering | **46.21** | **62.82** | **45.32** | **61.64** | **53.06** |
| 50% | SlimGPT w/o | 21.55 | 20.51 | 20.55 | 21.93 | 21.22 |
| | FLAP | 24.35 | 21.70 | **22.23** | **23.94** | 23.20 |
| | Token Filtering | **24.27** | **22.23** | 22.01 | 23.85 | **23.22** |

Table 15: MMLU zero-shot performance of Phi-4-14B. Here, "social" denotes the social sciences category. Probe Pruning (PP) is excluded since its official implementation does not support MMLU evaluation. A dash $(-)$ indicates that the baseline method could not be applied successfully to the corresponding model.

| Method | Pruning Ratio | Phi-4-14B | | | | |
| --- | --- | --- | --- | --- | --- | --- |
| | | Humanities | Social | STEM | Other | Avg |
| 0% | Dense | 39.21 | 45.99 | 33.17 | 45.41 | 40.81 |
| 20% | SlimGPT w/o | 24.65 | 22.06 | 22.35 | 24.26 | 23.48 |
| | FLAP | - | - | - | - | - |
| | Token Filtering | **63.40** | **82.09** | **63.34** | **76.86** | **70.46** |
| 50% | SlimGPT w/o | 24.18 | 21.70 | 21.34 | 23.97 | 22.95 |
| | FLAP | - | - | - | - | - |
| | Token Filtering | **29.69** | **34.68** | **31.43** | **38.20** | **33.06** |

## B  HYPERPARAMETER

In this section, we analyze the sensitivity of Token Filtering to its two primary hyperparameters: (1) the tail-layer ratio Y, which determines the fraction of layers eligible for pruning, and (2) the anchor smoothing factor $\gamma$, which controls how quickly the anchor key/value adapts to new tokens.

Table 16 reports the perplexity (PPL) and commonsense reasoning accuracy (ACC) of LLaMA-2-13B under a 33% pruning ratio while varying the tail-layer ratio $Y \in \{0.4, 0.5, 0.6\}$. The results indicate that Token Filtering is fairly robust to the choice of $Y$: both PPL and ACC remain stable for $Y = 0.4$ and $Y = 0.5$, whereas a larger value ($Y = 0.6$) leads to a small degradation. This indicates that smaller values of $Y$, which place a stronger focus on pruning the tail layers, lead to better overall performance.

Table 16: Perplexity (PPL) and commonsense reasoning accuracy (ACC) of LLaMA-2-13B under different values of the tail-layer ratio hyperparameter Y at a 33% pruning ratio

| Prune% | Y | PPL | ACC |
| --- | --- | --- | --- |
| 33% | Y = 0.4 | 16.39 | 68.81 |
| | Y = 0.5 | 16.39 | 67.65 |
| | Y = 0.6 | 18.86 | 62.75 |

Table 17 examines the effect of the anchor smoothing factor $\gamma$ under the same 33% pruning ratio. We observe that values in the range $\gamma \in [0.8, 0.9, 0.95]$ produce nearly identical results, while lowering the factor to $\gamma = 0.8$ results in a slight increase in perplexity. Since $\gamma$ determines how strongly the anchor retains historical key/value information, these results confirm that the method is not overly sensitive to this hyperparameter as long as $\gamma$ remains close to 1.0.

Table 17: Perplexity (PPL) and commonsense reasoning accuracy (ACC) of LLaMA-2-13B under different values of the anchor hyperparameter $\gamma$ at a 33% pruning ratio

| Prune% | $\gamma$ | PPL | ACC |
|---|---|---|---|
| | $\gamma = 0.95$ | 16.39 | 67.49 |
| 33% | $\gamma = 0.9$ | 16.39 | 67.65 |
| | $\gamma = 0.8$ | 17.45 | 67.64 |

## C    COMPARED WITH THE TOKEN PRUNING METHOD

In this section, we provide an additional comparison between Token Filtering and H2O (Zhenyu Zhang, 2023), a method designed for KV-cache eviction during long-context inference. Since Token Filtering and H2O operate in fundamentally different domains, the former prunes redundant tokens before attention computation, while the latter removes tokens from the KV cache after they have already been processed, a direct one-to-one comparison is not meaningful. Nevertheless, for completeness, we report results under a unified 50% KV-cache budget to illustrate how both approaches behave when constrained by the same memory limit.

Table 18 and 19 summarizes the accuracy, end-to-end throughput (token/s), and memory usage on LLaMA-2-13B. Under the matched 50% KV-cache budget, H2O achieves slightly higher accuracy (approximately +1 percentage point), reflecting its advantage in preserving more semantically important cached tokens. However, Token Filtering delivers higher throughput, achieving an 11% speedup over H2O due to skipping attention computation entirely for pruned tokens. Token Filtering shows slightly lower memory usage than H2O, but the difference is negligible in practice.

Overall, although the two techniques target different aspects of the inference pipeline and are therefore not directly comparable, this experiment demonstrates that Token Filtering remains competitive even when evaluated under a constrained KV-cache setting.

Table 18: Comparison of accuracy, end-to-end throughput (token/s), and memory usage on LLaMA-2-13B under a 50% KV-cache budget.

| KV cache budget% | Method | Accuracy(%) | TOPS(token/s) | Memory(MB) |
|---|---|---|---|---|
| 100% | Dense | 69.51 | 4.7 | 51531.9 |
| 50% | Token Filtering | 65.90 | 8.8 (+87%) | 38421.3 (-25.5%) |
| | H2O | 66.99 | 8.2 (+76%) | 38444.1 (-25.4%) |

Table 19: Zero-shot evaluation of LLaMA-2-13B on commonsense reasoning benchmarks. The "Avg." column reports the average accuracy across the seven tasks.

| KV cache budget% | Method | BoolQ | PIQA | HellaS | WinoG | ARC-e | ARC-c | OBQA | Avg. |
|---|---|---|---|---|---|---|---|---|---|
| 100% | Dense | 80.92 | 80.52 | 79.36 | 71.98 | 79.63 | 49.15 | 45.00 | 69.51 |
| 50% | Token Pruning | 79.76 | 77.04 | 74.56 | 71.03 | 71.72 | 45.22 | 42.00 | 65.90 |
| | H2O | 79.88 | 77.26 | 77.41 | 71.51 | 72.93 | 47.15 | 42.80 | 66.99 |

## D    GENERATIONS FROM PRUNED MODEL

To qualitatively examine the effect of pruning, we compared sentences generated by the LLaMA-2-13B model in its dense form and after applying 20% pruning. Table 20 presents the generations produced by LLaMA-2-13B in both the dense and 20% pruned settings.

| Model | Generated Sentences |
|---|---|
| **LLaMA-2-13B (baseline)** | In the future, artificial intelligence will be used to make decisions about our lives, from what we should buy and how we should spend our time, to what medical treatments we need. But there's a problem: it isn't very good at doing that. We're still learning how to do it right, so how can we improve? A new study published in Nature Communications suggests that AI could be able to solve some of its problems by using human intelligence as an input into the software. The researchers, led by the University of Oxford's Andrew Norton, analysed more than 10 million tweets from the US Twitter account @AI_Economist and found that the most common uses for AI were predicting the prices of goods on eBay, predicting stock market performance and predicting whether or not you have a disease. In all three cases, they were able to achieve better results with a ... |
| | Once upon a time, there was a little cat who loved to hide. She lived in a small house with her family, and they were all very happy together. One day, the little cat decided to explore the world outside of her home. She ventured out into the big wide world and found many new friends. Some of them were friendly, while others weren't so nice. But no matter what happened, the brave little cat always managed to find her way back home safely! This story is about a young girl named Little Cat, who loves to explore and play with her siblings. One day, she decides to go on an adventure by herself. Along the way, she meets some interesting people and animals that teach her important lessons about life. The moral of this story is that it's never too late to start something new or take risks if you want to grow as a person. The "i love little cat" is a children's book written by the ... |
| **LLaMA-2-13B (20% pruned)** | In the future, artificial intelligence will be able to perform tasks in almost all areas of life. The technology is already being used today in many different ways – for example, by Google's translation service or Amazon's voice assistant Alexa. In this post we explore why and how machine learning, the technical term for artificial intelligence, has established itself in so many applications. Where did the idea for AI come from? The idea that machines can think like humans was first expressed by the Greek mathematician Archimedes around 250 BC. He believed that if a machine could perform any mental task that a human could, then it would prove that mankind was not unique in the universe. More than two thousand years later, Charles Babbage built his "Difference Engine" as an early attempt at creating a mechanical calculator in the 19th century. However, his design was plagued with design ... |
| | Once upon a time, there was a little cat who wanted to be a big dog. He loved dogs and would try to act like one every chance he got. One day, he heard about a farm in the country where you could buy and sell animals—all kinds of animals! The thought of meeting so many different creatures made his tail wag with excitement. After hearing this news, he began thinking about how he could become a dog. The first thing he needed to do was find someone who could help him. So off he went looking for an animal doctor or veterinarian (or vet). As he roamed around town, he came across two very friendly people: Dr. Johnson and Nurse Betty! They were both very kind and happy to see him; they even offered him some delicious treats! But when they asked if he had come because he felt ill or injured, he admitted that wasn't why he'd stopped ... |

Table 20: Example generations from LLaMA-13B in the dense setting and with 20% pruning.

