# OpenReview forum: "KV-Prune: Key–Value Similarity for Online Structured Pruning for Large Language Models"
_ICLR.cc/2026/Conference — ICLR 2026 Conference Withdrawn Submission_

### Official Review · Reviewer_RYG1 · 2025-10-31

**Soundness:** 2
**Presentation:** 3
**Contribution:** 2
**Rating:** 4
**Confidence:** 3

**Summary:**

This paper introduces Token Filtering, a fully online structured pruning method for LLMs that accelerates inference without relying on global profiling or calibration data. The key idea is to measure token redundancy through the cosine similarity between current and historical key–value representations, allowing the model to skip redundant attention computations during decoding. To ensure stability, the method employs a variance-aware fusion strategy that adaptively weights key and value similarity across attention heads. It further adopts a tail-focused pruning scheme, applying pruning mainly to later layers with dynamically adjusted thresholds to minimize latency overhead. Experiments show its effectiveness and efficiency.

**Strengths:**

S1. [Originality and Clarity]: The paper proposes a novel and promising online pruning framework that removes the dependency on global profiling or calibration data. The overall methodology is easy to follow.

S2. [Experimental adequacy]: The experiments are comprehensive and convincing, covering multiple model scales (LLaMA-2/3, Mistral, Phi-4) and diverse benchmarks.

**Weaknesses:**

W1. [Inconsistent notation]: Some notations are inconsistent; for example, $alpha$ is used in Eq. (1)(2) and Eq. (4) with different meanings, which could create ambiguity.

W2. [Unclear implementation details]: The paper does not specify how residual connections and LayerNorm are handled when an attention layer is skipped—whether the attention output is zeroed out or bypassed directly.

W3. [Concerns about cold-start issue]: During the early steps of inference, when historical keys/values are insufficient, the similarity estimate may be unstable. The paper does not mention any warm-up or minimum-token safeguard. In addition, an ablation study on the threshold $\eta$ in threshold adaptation is necessary.

W4. [Confusing ablation naming]: Appendix tables use both “Key-Sim” and “Token Filtering” to describe variants, which can mislead readers about whether results correspond to the full model or ablations.

W5. [Overstated claims]: The claim of “no additional memory overhead” is inaccurate, since each layer must still store anchor and variance statistics. Therefore, a quantitative report of actual memory usage should be provided.

**Questions:**

I am particularly interested in the clarifications related to W2, W3, and W5 in WEAKNESSES, and I hope the authors can provide additional details on these aspects through the following questions:

Q1. How are residual connections and LayerNorm handled when an attention layer is skipped? Is the attention output zeroed out or directly bypassed?

Q2. How does the method address the cold-start issue when historical keys/values are insufficient? Is there any warm-up strategy or minimum context length before pruning is applied?

Q3. Could the authors provide quantitative evidence regarding the actual memory overhead introduced by storing anchor and variance statistics to support the claim of “no additional memory overhead”?

---

> ### Author Response · Authors · 2025-11-21
>
> We thank the reviewer for the insightful and constructive comments.
> We respond to all questions and concerns point-by-point below.
>
>
> ### **W1 & W4**
>
> **A:**
> Inconsistent notaion and confusing ablation naming have been changed.
> - $\alpha$ → $\gamma$ in Eq. (1)
> - $\alpha$ → $\eta$ in Eq. (4)
> - “Key-Sim” → “Token Filtering”
> ---
>
> ### **Q1: How are residual connections and LayerNorm handled when an attention layer is skipped? Is the attention output zeroed out or directly bypassed?**
>
> **A:**
> When an attention layer is skipped, LayerNorm is still computed, but the attention output is replaced with a zero vector. As a result, when the residual connection is added, the layer effectively behaves as an identity mapping.
>
> This explanation has been added to Section 3.2 of the revised paper.
>
>
>
> ---
>
> ### **Q2: How does the method address the cold-start issue when historical keys/values are insufficient? Is there any warm-up strategy or minimum context length before pruning is applied?**
>
> **A:**
> During the initial few tokens, when historical K/V are insufficient (the *cold-start* region), Token Filtering operates in **warm-up mode**, meaning it observes but does not perform any pruning. In this phase, the model only updates the anchor and learns the layer-wise thresholds.
>
> Each layer has a predefined **warm-up token count**, and pruning is enabled only after this minimum number of tokens has been processed.
> As a result, pruning never happens below this minimum context length, preventing any incorrect early-stage skipping.
>
> ---
> ### **Q3. Could the authors provide quantitative evidence regarding the actual memory overhead introduced by storing anchor and variance statistics to support the claim of “no additional memory overhead”?**
>
> **A:**
> Token Filtering maintains only one anchor key and one anchor value per layer, and both are stored per head. Under the tail-focused design, only half of the layers require anchors. Thus, the total anchor size is:
>
> $$
> (number\ of\ layers / 2) \times number\ of\ heads \times head\ dimenstion \times datatype\ size
> $$
>
> For LLaMA-13B in FP32, this corresponds to:
>
> $$
> (40/2) \times 40 \times 128 \times 4 \approx 409.6KB
> $$
>
>
> Anchor values have the same size, so the entire anchor memory footprint is approximately 800 KB, which is negligible compared to the KV cache, which is typically **~1.6 MB × batch size × sequence length.**
>
> In Token Filtering, the keys/values of pruned tokens are never stored in the KV cache, so the effective memory usage of the KV cache is reduced. In this context, the small anchor memory is insignificant relative to the large KV-cache memory.
>
> Finally, variance refers to per-token, per-head similarity variance, which is computed on-the-fly and not stored. Therefore, variance statistics introduce no additional memory overhead.
>
> These clarifications have been added to Section 3.2 of the revised paper.

---

### Official Review · Reviewer_Rp6S · 2025-10-31

**Soundness:** 3
**Presentation:** 3
**Contribution:** 2
**Rating:** 4
**Confidence:** 3

**Summary:**

This paper tackles the important problem of LLM inference latency by proposing Token Filtering, a novel online, zero-shot structured pruning method. Instead of relying on calibration data , it dynamically identifies redundant tokens during inference by measuring joint key-value (KV) similarity against the mean of past context. To improve stability, it uses a variance-aware fusion strategy that weights K and V similarity based on their consistency across heads. To minimize overhead, it employs a tail-focused pruning strategy, applying pruning only to the later, more redundant layers of the network. It studies LLaMA-2/3, Mistral, and Phi-4 models and demonstrates significant accuracy preservation and run-time inference speedups, especially at high pruning ratios.

**Strengths:**

The method is fully online and zero-shot, requiring no calibration dataset whatsoever. This simplifies deployment and avoids the generalization issues of offline pruning.

It demonstrates exceptional robustness at high pruning ratios (e.g., 50%). While baselines suffer from "severe model collapse," Token Filtering maintains strong performance, especially on complex tasks like MMLU .

**Weaknesses:**

The method's benefits are heavily skewed towards large batch sizes. At a small batch size (e.g., 8), the latency and memory reductions are modest (~12.5% and ~6%, respectively, per Figure 3).This is a significant limitation, as many real-world inference applications (like single-user chatbots) operate at a batch size of 1.

No Runtime Comparison to Baselines: The efficiency evaluation in Figure 3 only compares Token Filtering against the dense (unpruned) model. It lacks a direct runtime and latency comparison against the other pruning methods (SlimGPT, FLAP, PP) that are used in the accuracy tables. Without this comparison, it is impossible to evaluate the true efficiency-accuracy trade-off. For example, a baseline might have slightly lower accuracy but be significantly faster, which could be a preferable trade-off.

While task accuracy is consistently superior, the perplexity (PPL) is often slightly worse than the best baseline (Probe Pruning). For example, on LLaMA-2-13B at 50% pruning, Token Filtering's PPL is 29.22 vs. PP's 28.86.

The incremental averaging strategy for the anchor uses a fixed smoothing factor α=0.9, which seems heuristic. The paper does not include a sensitivity analysis for this hyperparameter.

**Questions:**

Listed as above in the weakness.

---

> ### Author Response · Authors · 2025-11-21
>
> We sincerely thank the reviewer for the clarifying and constructive comments. We carefully considered each suggestion and provide detailed responses below.
>
>
> ### **W1 & W2**
>
> **A:**
> The smaller performance gains observed at low batch sizes do not stem from an inherent limitation of Token Filtering, but rather from the structural characteristics of LLM inference itself. Even without pruning, when comparing LLaMA-13B and LLaMA-7B on the same GPU, the 7B model achieves about 40% higher throughput at batch size 1. However, at batch size 32, this gap widens to approximately 70%. This illustrates that as the batch size increases, the computational advantage of smaller models grows nonlinearly, which is a fundamental property of LLM inference.
>
> This trend is consistently observed across other pruning methods as well. In small-batch settings, the improvement achieved by any pruning technique is inherently limited, whereas differences between methods become more pronounced at larger batch sizes. Token Filtering follows the same general pattern. The following table reports the latency comparison between our method, the dense baseline, and other pruning methods.
>
>
> | 50% pruning | Method          | 8 batch | 16 batch | 32 batch | 64 batch | 128 batch |
> |-------------|-----------------|---------|----------|----------|----------|-----------|
> |             | Dense           | 16.8 t/s | 16.1 t/s | 11.8 t/s | 4.7 t/s  | 2.7 t/s   |
> |             | Token Filtering | 19.4 t/s | 17.8 t/s | 14.9 t/s | 8.8 t/s  | 5.2 t/s   |
> |             | FLAP            | 19.4 t/s | 19.1 t/s | 15.4 t/s | 9.6 t/s  | 5.8 t/s   |
> |             | Probe           | 19.2 t/s | 17.2 t/s | 14.6 t/s | 7.5 t/s  | 4.4 t/s   |
>
> This results has been added to Section 4.2.2 of the revised paper
>
> ---
>
> ### **W3**
>
> **A:**
> PPL is only marginally higher than Probe Pruning by about 0.36, whereas downstream task accuracy is up to 13.97% higher. Token Filtering also achieves lower latency compared to PP. In other words, while Token Filtering shows only a negligible increase in PPL, it outperforms Probe Pruning in both accuracy and latency.
>
> ---
>
> ### **W4**
>
> **A:**
> The hyperparameter originally used for computing the anchor, α, has been replaced with γ.
> The key idea of γ (for anchoring) is to maintain the influence of the recent window, so we use a fixed value close to 1.
> Since values of γ sufficiently close to 1 do not significantly affect model performance, the value γ = 0.9 was selected based on experimental results.
>
> | Prune% | γ       | PPL   | ACC   |
> |--------|---------|-------|-------|
> | 33%    | γ = 0.95 | 16.39 | 67.49 |
> | 33%    | γ = 0.9  | 16.39 | 67.65 |
> | 33%    | γ = 0.8  | 17.45 | 67.64 |

---

### Official Review · Reviewer_YrXu · 2025-10-31

**Soundness:** 2
**Presentation:** 3
**Contribution:** 2
**Rating:** 4
**Confidence:** 4

**Summary:**

This paper proposes Token Filtering, a method to dynamically skip attention computations on tokens that have high cosine similarity to the average key/value in the preceding tokens. The method achieves higher accuracy than the compared baselines while reducing latency and memory overhead, particularly at large batch sizes.

**Strengths:**

* The proposed method helps reduce the quadratic complexity of the attention module, a source of significant overhead in LLM inference.
* Token Filtering is calibration-free, ensuring general applicability across a broad range of contexts.
* Token Filtering achieves high accuracy compared to the baselines and suffers less degradation at higher compression rates in particular.
* The variance-aware metric helps ensure that token representations that are particularly different within specific attention heads remain unpruned.
* The paper includes ablations on which layers to prune and the similarity metric.

**Weaknesses:**

# Major concerns
The following represent key weaknesses that must be addressed to increase the rating:
* Support for static computational graph: Dynamic conditional computation is challenging to integrate with modern JIT compilers, which generally require a static computational graph. Prior work such as [1] incorporates a top-k routing approach to ensure that exactly k tokens are routed through the conditional branches. Can such a scheme be incorporated into Token Filtering? And if so, how does top-K routing affect the accuracy and performance of the Token Filtering? [1] is a closely related work that should be cited in this paper.
* K/V anchor value causality: The paper primarily refers to using Token Filtering in the decoding stage, it’s a little ambiguous whether the method can be applied to prefill as well. If it can be, how the anchor values are determined during prefill is unclear. Are the K/V anchor values averaged over the entire input sequence or only on the tokens prior to the current query token?
* Token pruning baselines: The baselines include layer-pruning and structured pruning  (neuron/channel) methods; however, Token Filtering shares some features with Token Pruning methods such as LazyLLM. While I understand the authors argument that their method is more closely related to structured pruning / conditional computation methods, a direct comparison with an established token pruning method is crucial to better understanding the accuracy/latency trade-off between these approaches.
* Long-context performance: The quantitative evaluations are conducted on standard downstream QA tasks or 128 token long sequences of WikiText. These are relatively short prompts. Whether Token Filtering retains its high accuracy at long contexts remains to be evaluated. In particular, tasks from RULER such as Needle-in-a-Haystack may represent a significant challenge for Token Filtering when relying on K/V similarity.
* Overhead quantification: While the tail-focused strategy is introduced as a way to mitigate the overhead of online similarity computations, quantification of the overhead is not provided.
* Hyperparameter tuning: The method introduces new hyperparameters such as $Y$ and two $\alpha$ variables. It’s unclear whether these parameters required tuning and how sensitive Token Filtering is to them.
* Additional benchmarking information: The benchmarking settings are lacking some important details such as the sequence length, time-to-first-token (TTFT), and tokens-output-per-second (TOPS). Based on the increasing latency of attention w.r.t. batch size, it appears that the benchmark setting may be a single input with increasing input sequence length?

# Minor concerns
The following are minor concerns, typos, etc. which would improve the work but do not affect the final rating:
* PPL != Text Generation: Table 2 and associated text introduce WikiText PPL as indicative of text generation. In general, PPL can be a misleading metric when evaluating compressed LLMs and language modelling metrics such as PPL are not necessarily strong proxies for open-ended generative tasks.
* Extension to hybrid attention models: Many modern open-weight LLMs employ hybrid attention with interleaved sparse (local) / dense attention. Token Filtering is not evaluated on such architectures.

[1] D. Raposo, S. Ritter, B. Richards, T. Lillicrap, P. C. Humphreys, and A. Santoro, “Mixture-of-Depths: Dynamically allocating compute in transformer-based language models,” Apr. 02, 2024, arXiv: arXiv:2404.02258. [Online]. Available: http://arxiv.org/abs/2404.02258

**Questions:**

* Can TokenFiltering be applied to prefill? How are the K/V anchor values determined in this setting?
* How does LazyLLM compare with Token Filtering in terms of accuracy, latency, and memory?
* How does Token Filtering compare to baseline on RULER or other long-context benchmarks?
* What is the overhead of Token Filtering when 50% of tokens are skipped across a given decoder? What is the overhead when no tokens are skipped?
* How sensitive is Token Filtering to $Y$? What about $\alpha$ for anchoring and $\alpha$ for calculating $T_l$? How were the selected values of 0.5, and 0.9 determined? What value is used for $\alpha$ in the $T_l$ update expression?
* What input sequence length is used for the benchmark results? Is the latency measured across both prefill and decoding or only on one phase? What is the TTFT and TOPS for the various batch sizes? What did the input consist of?
* Given the dynamic nature of Token Filtering, some inputs are likely to include more tokens with similarity exceeding $T_l$. Is $T_l$ calculated on a per input basis or a global average across all prior inputs? Empirically, what is the mean and variance of the number of tokens filtered across a range of typical inputs?

---

> ### Author Response · Authors · 2025-11-21
> **Official Comment by Authors (1)**
>
> We deeply appreciate the reviewer’s careful reading and insightful comments. The raised concerns were highly valuable and helped us substantially improve the paper.
>
> ### **Q1: Can TokenFiltering be applied to prefill? How are the K/V anchor values determined in this setting?**
>
> **A:**
> Token Filtering employs a warm-up phase in which the anchor is updated to preserve model accuracy, but no pruning is performed. The warm-up length (i.e., the number of warm-up tokens) is fully configurable by the user. In an extreme setting, if the warm-up period is set very short, pruning could occur even during the prefill stage; however, this is not recommended, and in all of our experiments, no pruning was applied during prefill.
>
> This clarification has been added to Section 3.2 of the revised paper.
>
> ---
>
> ### **Q2: How does LazyLLM compare with Token Filtering in terms of accuracy, latency, and memory?**
>
> **A:**
> LazyLLM focuses primarily on optimizing the prefill stage, which is why we compared it against H2O. However, since Token Filtering and H2O operate in different domains, a direct comparison is not strictly meaningful. Therefore, we evaluate them under their largest shared constraint: a fixed KV-cache size budget.
>
> We evaluate both Token Filtering and H2O under a 50% KV-cache budget using LLaMA-2-13B with a batch size of 64. H2O shows approximately 1 percentage point higher accuracy, but Token Filtering achieves ~11% higher throughput (TOPS). Memory usage is slightly lower for Token Filtering, though the difference is small enough to be considered negligible.
>
> | Method           | Accuracy (%) | TOPS (token/s)   | Memory (MB)       |
> |------------------|--------------|------------------|--------------------|
> | Dense            | 69.51        | 4.7              | 51531.9            |
> | Token Filtering  | 65.90        | 8.8 (+87%)       | 38421.3 (-25.5%)   |
> | H2O              | 66.99        | 8.2 (+76%)       | 38444.1 (-25.4%)   |
>
> This result has been added to Appendix C of the revised paper.
>
> ---
>
> ### **Q3: How does Token Filtering compare to baseline on RULER or other long-context benchmarks?**
>
> **A:**
> We evaluate Needle-in-a-Haystack (NiH) performance across six context lengths: 512, 1024, 1536, 2048, 2560, and 3072 tokens. For each context length, we insert the needle at 20 different depth positions (from 1% to 99% of the document), and measure the retrieval success rate.
>
> | Prune% | Method         | 512  | 1024 | 1536 | 2048 | 2560 | 3072 |
> |--------|----------------|------|------|------|------|------|------|
> | 0%     | Dense          | 100% | 100% | 90%  | 95%  | 45%  | 35%  |
> | 20%    | Token Filtering| 70%  | 70%  | 65%  | 55%  | 30%  | 10%  |
> | 20%    | FLAP           | 25%  | 20%  | 10%  | 0%   | 0%   | 0%   |
>
> This result has been added to Section 4.3.3 of the revised paper.
>
> ---
>
> ### **Q4: What is the overhead of Token Filtering when 50% of tokens are skipped across a given decoder? What is the overhead when no tokens are skipped?**
>
> **A:**
> The latency-overhead experiment at batch size 64 shows the following results.
> Under 50% pruning, the Token Filtering layer introduces only 0.32 ms of latency, while the attention latency is reduced by more than half, resulting in a total latency reduction of 5.92 ms. Furthermore, in the official experimental setup of the paper, Token Filtering is deactivated when no pruning occurs, meaning that it introduces no additional overhead during those steps.
>
> |                | Attention layer | Token Filtering layer | Total    |
> |----------------|-----------------|------------------------|----------|
> | Base           | 11.51 ms        | 0 ms                   | 12.24 ms |
> | 50% filtering  | 5.33 ms         | 0.32 ms                | 6.32 ms  |

---

> ### Author Response · Authors · 2025-11-21
> **Official Comment by Authors (2)**
>
> ### **Q5: How sensitive is Token Filtering to Y ? What about α for anchoring and α for calculating Tl ? How were the selected values of 0.5, and 0.9 determined? What value is used for α in the Tl update expression?**
>
> **A:**
> As shown in the table below, a smaller value of Y causes more pruning to be concentrated in the later layers, which generally improves both perplexity (PPL) and accuracy (ACC). However, a very small Y makes it difficult to achieve high pruning ratios. Since our paper evaluates pruning ratios up to 50%, we chose Y = 0.5.
>
> | Prune% | Y  | PPL  | ACC   |
> |--------|--------|------|-------|
> | 33%    | Y = 0.4| 16.39| 68.81 |
> | 33%    | Y = 0.5| 16.39| 67.65 |
> | 33%    | Y = 0.6| 18.86| 62.75 |
>
>
> The hyperparameter originally used for computing the anchor, α, has been replaced with γ.
> The key idea of γ (for anchoring) is to maintain the influence of the recent window, so we use a fixed value close to 1.
> Since values of γ sufficiently close to 1 do not significantly affect model performance, the value γ = 0.9 was selected based on experimental results.
>
> | Prune% | γ       | PPL   | ACC   |
> |--------|---------|-------|-------|
> | 33%    | γ = 0.95 | 16.39 | 67.49 |
> | 33%    | γ = 0.9  | 16.39 | 67.65 |
> | 33%    | γ = 0.8  | 17.45 | 67.64 |
>
> The parameter α used to compute $T_l$ has been replaced with η, and the paper uses η = 1. We found that η functions correctly and does not affect performance as long as it is not set to an extremely large or small value. This is because the similarity distribution of tokens is discrete; therefore, even if the threshold changes slightly, the filtering result remains unchanged unless the set of filtered tokens itself changes. Consequently, η has little impact on the final performance.
>
> This result has been added to Appendix B of the revised paper.
>
> ---
>
> ### **Q6: What input sequence length is used for the benchmark results? Is the latency measured across both prefill and decoding or only on one phase? What is the TTFT and TOPS for the various batch sizes? What did the input consist of?**
>
> **A:**
> All latency measurements were performed using the prompt “Once upon a time in a land far, far away,” which corresponds to 13 tokens under the LLaMA tokenizer. The reported latency reflects end-to-end inference time; however, because the input sequence is very short, the contribution of the prefill phase is minimal, and most of the performance improvement arises during the decode stage.
>
> We also used the prompt "Once upon a time in a land far, far away" for the TTFT and TOPS experiments. For TOPS, pruning improved throughput by 87% at batch size 64 and 92% at batch size 128, which aligns well with the results shown in Figure 3 of the paper.
>
> For TTFT, the trends differ: at all batch sizes, the dense model is about 15% faster than the 50% pruned model.This is because Token Filtering performs additional computations during the prefill phase, such as anchor updates, which slightly increase the TTFT.
>
> |       Metric       | Method         | 8        | 16       | 32       | 64       | 128      |
> |--------------------|----------------|----------|----------|----------|----------|----------|
> | **TOPS**           | Dense          | 16.8 t/s | 16.1 t/s | 11.8 t/s | 4.7 t/s  | 2.7 t/s  |
> |                    | 50% pruning    | 19.4 t/s | 17.8 t/s | 14.9 t/s | 8.8 t/s  | 5.2 t/s  |
> | **TTFT**           | Dense          | 0.35 s   | 0.37 s   | 0.42 s   | 0.48 s   | 0.55 s   |
> |                    | 50% pruning    | 0.42 s   | 0.44 s   | 0.47 s   | 0.56 s   | 0.61 s   |
>
> This result has been added to Section 4.2.2 of the revised paper.
>
> ---
>
> ### **Q7: Given the dynamic nature of Token Filtering, some inputs are likely to include more tokens with similarity exceeding $T_l$. Is $T_l$ calculated on a per input basis or a global average across all prior inputs? Empirically, what is the mean and variance of the number of tokens filtered across a range of typical inputs?**
>
> **A:**
> $T_l$ is determined by the difference between the user-specified target skip ratio and the current skip ratio.
> Similar to gradient descent, if the current skip ratio becomes higher than the target skip ratio, $T_l$ increases to reduce the amount of skipping (because tokens with similarity above $T_l$ are skipped; therefore, a lower $T_l$ results in more skipping, while a higher $T_l$ results in less skipping).
> Further details are provided in Section 3.3 Tail-Focused Pruning with Layer-Wise Thresholds.
>
> In general, pruning methods allow users to specify a target pruning ratio, and Token Filtering follows the same principle.
> Specifically, the user can choose what percentage of tokens should be filtered, effectively controlling the pruning ratio applied during inference.

---

### Note · Authors · 2026-02-04

I have read and agree with the venue's withdrawal policy on behalf of myself and my co-authors.

---

### Meta-Review · Area_Chair_McDZ · 2026-01-06

**Summary:**

Reviewers noted the following concerns:
- Whether the proposed scheme can be applied to prefill for more practical deployment
- Experimental settings lacked key details
- Quite a number of hyperparameters
- More concrete quantification of overhead.

**Reviewer Concerns:**

Applicability to prefill setting is largely addressed.

Others are partially or not addressed.
- hyperparameters: only a brief experiment on one of the hyperparameters with limited range of values
- quantification of memory overhead - only analytical discussion, no actual memory overhead experiment.

**Reviewer Scores:**

Reviewer YrXu - might have maintained same rating.
Others - might have increase by 0-1

---

### Decision · Program_Chairs · 2026-01-26

Reject